# MPP: MODEL POWER PARITY FOR REASONING LARGE LANGUAGE MODEL COMPARISON ON CROSS-DOMAIN BENCHMARKS

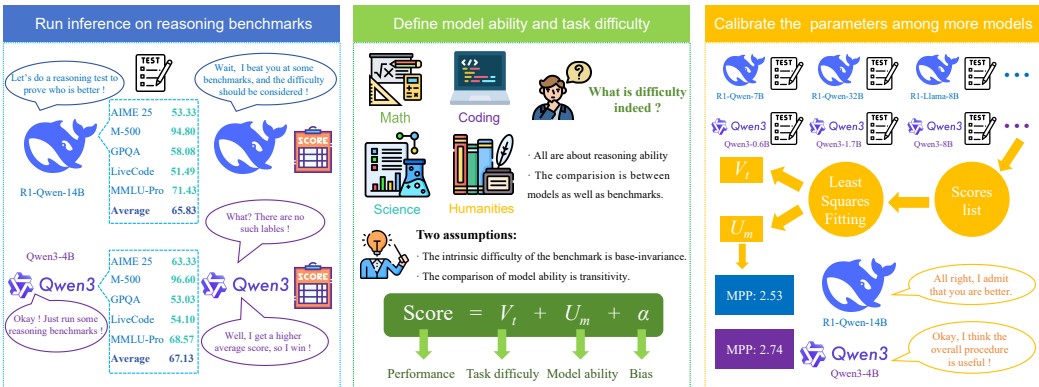

Figure 1: Schematic diagram of our model ability comparison framework. When comparing model capabilities, the scores of models on various benchmarks are usually directly averaged, and the average scores are compared. However, as illustrated in the figure, there are cases where one model cannot outperform another across all benchmarks, leading to insufficient persuasiveness of the comparison results. To address this issue, we propose a concise linear regression model that decomposes model scores into two dimensions: **task difficulty and model capability**. We fit this model on the score results of dozens of models, enabling cross-domain comparison of model capabilities.

## ABSTRACT

The ability of Large Language Models is usually evaluated by their average performance scores on various benchmarks. However, this way of comparison does not consider the difficulty difference between benchmarks, and the average score also lacks interpretability. Inspired by the International Comparison Program (ICP), we introduce model power parity (MPP), a principled framework that adapts the idea of purchasing power parity (PPP) to LLM evaluation. MPP performs multilateral, direct comparisons among models and benchmarks without assuming a common difficulty scale. Extensive experiments show that the MPP framework provides reasonable evaluation results and brings new insight into the relation between models and benchmarks. We evaluate over 30 open-source reasoning large language models on 8 prevalent reasoning benchmarks. Our results reveal three key insights: (1) The idea of PPP suits well for building a ranking framework evaluating models' ability and benchmark difficulty. (2) MPP manages to infer missing performance scores matching actual inference results within about 0.5 to 3 points on top-2 benchmarks. (3) MPP helps distinguish whether a benchmark is suitable for evaluating a given model family. We believe that MPP offers a fresh, economically grounded perspective on equitable LLM capability assessment and will facilitate more reliable model selection and benchmark building.

Table 1: Model Power Parity (MPP) and average scores on 8 reasoning benchmarks of 16 models. A greater value represents a more powerful reasoning ability, and "awq-4" denotes the model that adopts the AWQ quantization method (Lin et al., 2024) with 4-bit precision. The models with colored scores exhibit changes in their capability rankings under the two evaluation methods.

| Model | MPP↑ | Avg↑ | Model | MPP↑ | Avg↑ |
|---|---|---|---|---|---|
| Qwen3-32B | 4.21 | 73.67 | DeepSeek-R1-Distill-Qwen-14B | 2.53 | 64.42 |
| QwQ-32B | 4.20 | 73.l7 | Phi-4-mini-reasoning | 1.54 | 53.59 |
| Qwen3-14B | 3.79 | 71.82 | DeepSeek-R1-Distill-Qwen-7B | 1.47 | 53.76 |
| Qwen3-8B | 3.27 | 68.84 | Qwen3-1.7B | 1.25 | 50.58 |
| Phi-4-reasoning | 3.19 | 69.37 | DeepSeek-R1-Distill-Llama-8B | 1.13 | 49.41 |
| DeepSeek-R1-Distill-Llama-70B | 2.86 | 66.66 | DeepSeek-R1-Distill-Qwen-1.5B | 0.62 | 38.03 |
| DeepSeek-R1-Distill-Qwen-32B | 2.85 | 66.19 | Qwen3-0.6B | 0.39 | 32.33 |
| Qwen3-4B | 2.74 | 64.92 | Qwen3-0.6B-awq-w4 | 0.26 | 26.16 |

# 1 INTRODUCTION

Large language models (LLMs) have undergone rapid advancement in recent years, accompanied by the rapid increase of benchmarks across diverse domains, such as:

- General knowledge (Wang et al., 2024; Suzgun et al., 2023; Hendrycks et al., 2021b),
- Common-sense reasoning (Zellers et al., 2019; Clark et al., 2018; Mihaylov et al., 2018),
- Mathematics (Lightman et al., 2023; Hendrycks et al., 2021c; Cobbe et al., 2021),
- Coding (Jain et al., 2024; Chen et al., 2021; Hendrycks et al., 2021a).

When assessing LLMs with varying parameter sizes and architectural designs, some models may excel on certain benchmarks yet underperform relative to others on different ones, creating significant obstacles to objectively comparing the overall capabilities of different LLMs. A common practice to address this is to directly average a model's scores across all benchmarks and compare the mean values. This approach, however, **overlooks critical differences** in benchmark difficulty. Such differences exist not only across domains but also within the same domain. For example, mathematics benchmarks may include both middle school-level questions and university-level math competition problems. Simply averaging every score not only **lacks explainability** but may also produce biased rankings, which affects the objective evaluation of the model's ability.

Existing efforts to estimate benchmark difficulty still have notable limitations. Some benchmarks (Rein et al., 2023; Bean et al., 2024; Shoham & Rappoport, 2024; Yang et al., 2024; Ding et al., 2024) attach difficulty-related labels to questions or question pairs. These labels may take the form of specific scores, discrete levels, relative relationships, or quantitative metrics, and they usually involve human experts' participation or draw on domain-specific rules. The difficulty definitions of these benchmarks are mainly tailored to their own scenarios and fail to establish a universal cross-domain difficulty standard. Recent works like Easy2Hard-Bench (Ding et al., 2024) propose a collection of 6 cross-domain datasets with standardized difficulty labels for LLM assessment and provide new insights for benchmark difficulty estimation, but it is time-consuming to scale up or estimate difficulty for new benchmarks. Moreover, existing works do not explicitly qualify the relationship between a model's ability and benchmark difficulty, and this oversight is notable given that such a relationship is critical for rigorous comparisons among models and benchmarks.

The International Comparison Program (ICP) is one of the largest and most enduring statistical initiatives in the world (World Bank, 2025). This initiative has two primary objectives: first, to generate purchasing power parities (PPPs) and comparable price level indexes (PLIs) for all economies involved; and second, to translate volume-based and per capita metrics of gross domestic product (GDP) as well as its expenditure components into a common currency by leveraging PPPs. The purchasing power parity (PPP) between the currencies of countries A and B can be defined as the number of currency units of country A that have the same purchasing power as one unit of the currency of country B. Based on this concept, we draw an analogy: just as different countries use different currencies whose values cannot be directly compared without considering price levels, different LLMs produce performance scores on different benchmarks that are not directly comparable due to varying task difficulty. In the same way that a currency with higher purchasing power can

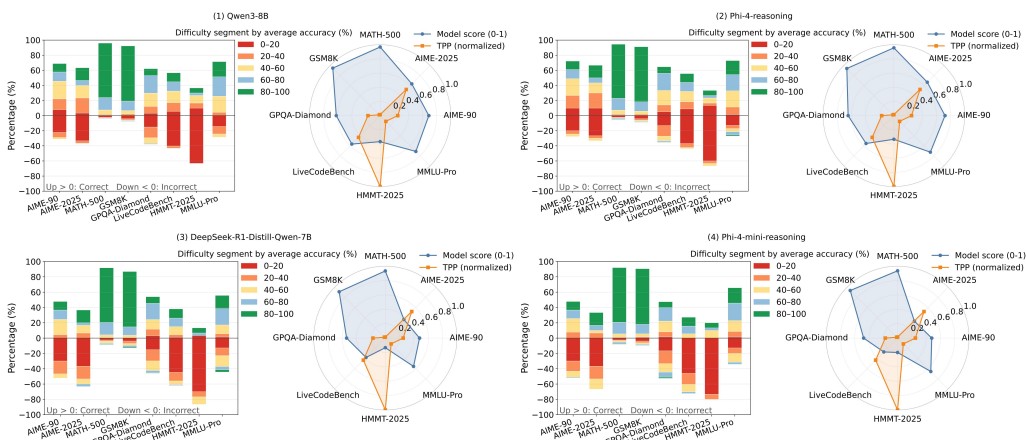

Figure 2: This figure presents the scores of four models (Deepseek-R1-Distill-Qwen-7B, Qwen3-8B, Phi-4-reasoning, and Phi-4-mini-reasoning) across 8 reasoning benchmarks, as well as their answer accuracy rates across different question difficulty tiers. In the bar charts, each question from all benchmarks is categorized into 5 difficulty levels based on the overall answer accuracy rates of all models, and the proportion of questions corresponding to each difficulty level within each benchmark is statistically summarized. In the radar charts, we simultaneously display the models' scores on each benchmark and the TPP values of each benchmark.

buy more goods in different countries, a stronger model can "purchase" higher scores across benchmarks with different difficulty levels. We present an intuitive visualization of the entire workflow for our proposed comparison framework in Figure 1. Building on the computational framework of PPP, we propose the MPP framework, which we use to conduct reasonable comparisons across over 30 advanced open source reasoning models, including Qwen3 (Yang et al., 2025), DeepSeek-R1 (Guo et al., 2025), Phi-4 (Xu et al., 2025; Abdin et al., 2025), and QwQ (Qwen Team, 2025). Our contributions in this paper can be summarized as below:

**MPP framework:** We propose a mathematical framework that simultaneously estimates model ability and benchmark difficulty. This model avoids the tedious benchmark difficulty annotation and addresses the limitation of simple average score calculation, which fails to account for the difficulty differences among benchmarks.

**Performance prediction:** We find that the MPP framework can infer missing performance without additional inference, matching actual results within about 0.5 to 3 points on top-2 benchmarks.

**Benchmark selection:** We derive a data-driven criterion to decide whether a benchmark is appropriate for evaluating a given model family.

## 2 METHODOLOGY

### 2.1 BASIC CONCEPT OF PURCHASING POWER PARITY

The PPP theory in this section mainly refers to the release of the World Bank (Rao, 2013).

Purchasing Power Parity (PPP) is the exchange rate that equalizes the purchasing power of two currencies, such that a given amount of currency in country A can acquire the same basket of goods or services as one unit of currency in country B, typically defined at the level of a **basic heading**. Suppose $p_{ij}$ and $p_{ik}$ represent the prices of product $i$ in countries $j$ and $k$, respectively. The Purchasing Power Parity (PPP) for country $k$ with respect to country $j$ is then given by:

$$PPP_{jk} = \frac{p_i^k}{p_i^j}. \tag{1}$$

For a given commodity $i$, the following transitivity property can be observed. Specifically, for any three countries $j$, $k$, and $m$, if th,e PPP between countries $j$ and $k$, as well as between countries $k$ and $m$, is known, it is straightforward to conclude that:

$$PPP_{jk} = \frac{p_i^k}{p_i^j} = \frac{p_i^k}{p_i^m} \times \frac{p_i^m}{p_i^j} = PPP_{jm} \times PPP_{mk}. \tag{2}$$

**Transitivity**: Multilateral Purchasing Power Parities (PPPs), represented by the matrix of PPP comparisons between all pairs of countries based on price data from multiple items, are considered **transitive** For any three countries in the group, such as $j$, $k$, and $m$, the direct PPP between country $k$ and country $j$ is equal to the indirect PPP calculated through the use of the third country, $m$:

$$PPP_{jk} = PPP_{jm} \times PPP_{mk} = \frac{PPP_{mk}}{PPP_{mj}}. \tag{3}$$

**Base Invariance**: It is essential that all countries involved in the price comparisons are treated symmetrically without assigning special status to any particular country, which is called base-invariance.

## 2.2 COUNTRY PRODUCT DUMMY METHOD

At the basic heading level, the two primary aggregation methods are the Country Product Dummy (CPD) method and the Gini-Éltető-Köves-Szulc (GEKS) method. We mainly focus on the CPD method, and the details about the GEKS method can be found in the Appendix B.1. The CPD method is a regression-based approach used to calculate PPP between countries. Let $p_{ij}$ denote the price of item $i$ in country $j$. The CPD model can be expressed as:

$$p_{ij} = PPP_j \cdot P_i \cdot u_{ij} \quad \text{for } j = 1, 2, \ldots, C; \; i = 1, 2, \ldots, N, \tag{4}$$

Where $PPP_j$ denotes the purchasing power parity of the currency of country $j$ relative to a reference country, and thus one subscript is omitted. $P_i$ represents the international average price of the $i$-th commodity, and $u_{ij}$ are independently and identically distributed disturbance terms. Within this framework, the CPD model estimates country-specific price levels ($\alpha_j$) via regression, thereby reflecting relative purchasing power. Interpreted more generally as a hedonic regression, it incorporates country and commodity effects, and in **logarithmic form** can be expressed as:

$$\ln(p_{ij}) = \ln(PPP_j) + \ln(P_i) + \ln(u_{ij}) = \alpha_i + \gamma_i + \nu_{ij}, \tag{5}$$

Where $\nu_{ij}$ denotes independently and identically distributed random disturbances with zero mean and constant variance $\sigma^2$, and $\alpha_j$ is interpreted as the purchasing power parity of country $j$, measuring the number of currency units in country $j$ that are equivalent in purchasing power to one unit of the **reference country's** currency. These parameters can be estimated using simple ordinary least squares, and the details can be found in the Appendix B.2. Accordingly, the PPP for country $j$ is estimated as:

$$PPP_j = \exp(\hat{\alpha}_j). \tag{6}$$

Since these PPPs are derived from estimated parameters, their associated standard errors can also be computed, thereby providing a measure of statistical precision. The estimated variance of $PPP_{jk}$ is given by:

$$EstVar(\hat{\alpha}_j) = \frac{2}{N}\hat{\sigma}^2, \tag{7}$$

where $\hat{\sigma}^2$ is an unbiased estimator of $\sigma^2$, which is given by:

$$\hat{\sigma}^2 = \frac{\sum_{j=1}^{C} \sum_{i=1}^{N} e_{ij}^2}{CN - (C + N - 1)}, \tag{8}$$

Where $e_{ij} = \ln p_{ij} - \hat{\alpha}_j - \hat{\gamma}_i$ is the least squares residual. The estimated variance of $PPP_j$ with a numeraire country is given by:

$$EstVar(\hat{PPP_j}) \approx EstVar(\hat{\alpha}_j) \cdot (\hat{\alpha}_j)^2. \tag{9}$$

Equation 9 can then be used in deriving the estimated variance for PPPs with any other country as the reference country.

Table 2: Models' performance on 8 benchmarks with predicted scores. The values in parentheses represent predicted performance, while the values without parentheses are actual performance. Entries that include both predicted and actual performance represent the assumption that the score is missing and will be predicted. For example, "85.60 (85.16)" means that the actual performance is 85.60 and the predicted performance is 85.16. The full score is 100 for all benchmarks.

| Model | AIME-90 | AIME 25 | MATH-500 | GSM8K | GPQA-Diamond | LiveCodeBench | HMMT 25 Feb | MMLU-Pro |
|---|---|---|---|---|---|---|---|---|
| DeepSeek-R1-Distill-Qwen-1.5B | 22.22 | 20.00 | 85.60 (85.16) | 83.78 (84.27) | 34.85 | 14.93 | 10.00 | 32.86 |
| DeepSeek-R1-Distill-Qwen-7B | 47.78 | 36.67 | 93.40 (93.13) | 91.13 | 54.04 (55.24) | 38.06 | 13.33 | 55.71 |
| DeepSeek-R1-Distill-Llama-8B | 42.22 | 26.67 | 88.40 | 88.63 | 40.91 | 36.57 | 13.33 (13.31) | 58.57 (59.14) |
| DeepSeek-R1-Distill-Qwen-14B | 62.22 (60.11) | 53.33 | 94.80 (96.03) | 94.01 | 58.08 | 51.49 | 30.00 | 71.43 |
| DeepSeek-R1-Distill-Qwen-32B | 63.33 | 56.67 | 96.60 (96.32) | 94.24 | 64.14 | 57.84 | 26.67 (27.77) | 70.00 |
| QwQ-32B | 80.00 | 73.33 | 97.40 (97.50) | 95.45 | 63.13 | 60.82 | 36.67 (36.25) | 78.57 |
| DeepSeek-R1-Distill-Llama-70B | 64.44 (62.89) | 53.33 | 95.60 (96.44) | 94.01 | 65.15 | 55.97 | 23.33 | 81.43 |
| Qwen3-0.6B | 12.22 | 3.33 | 72.60 | 78.32 (76.33) | 29.29 | 13.81 | 13.33 | 35.71 (32.26) |
| Qwen3-1.7B | 38.89 | 33.33 | 91.20 (92.02) | 89.76 | 37.37 | 31.72 | 16.67 (14.30) | 65.71 |
| Qwen3-4B | 58.89 | 63.33 | 96.60 (96.28) | 94.84 (96.06) | 53.03 | 54.10 | 30.00 | 68.57 |
| Qwen3-8B | 68.89 | 63.33 | 96.80 (97.02) | 94.77 (96.85) | 62.12 | 56.72 | 36.67 | 71.43 |
| Qwen3-14B | 71.11 | 70.00 | 96.60 (97.50) | 95.75 (97.33) | 64.14 | 62.69 | 40.00 | 74.29 |
| Qwen3-32B | 80.00 | 70.00 | 97.20 (97.78) | 95.30 (97.66) | 69.19 | 61.94 | 40.00 | 75.71 |
| Phi-4-mini-reasoning | 47.78 | 33.33 | 93.60 (93.43) | 93.63 | 47.47 | 27.24 | 20.00 | 65.71 (66.22) |
| Phi-4-reasoning | 72.22 | 66.67 | 95.60 (96.84) | 94.01 | 64.65 | 55.60 | 33.33 (30.62) | 72.86 |

## 2.3 CPD-MTD FORMULA

To facilitate the calculation of the regression model, we first normalize and clip the raw scores, which fall within the range $[0, 100]$ to value $p$, convert them to odds, and finally take the natural logarithm; for perplexity (PPL), we simply take the negative logarithm. This transformation smooths the raw scores, centers the zero point at a 50% accuracy rate, and converts multiplicative relationships into additive ones, maintaining mathematical consistency with the original CPD formula:

$$\text{score} = \begin{cases} \ln\left(\dfrac{p}{1-p}\right), & \epsilon \le p \le 1 - \epsilon, \epsilon = 1e-5 \\ -\ln(\text{PPL}), & \text{PPL} > 0. \end{cases} \tag{10}$$

We argue that a model's final performance on a benchmark is determined by both the model's inherent capability and the intrinsic difficulty of the benchmark. **Referring to Formula 5, we analogize models to currencies and benchmarks to commodities, which is analogous to how a model's capability enables it to "purchase" scores on each benchmark.** Accordingly, we propose the mathematical model describing the relation between them:

$$\text{score} = \alpha + U_m + V_t. \tag{11}$$

We refer to Formula 11 as the CPD method for calculating M models' ability and the difficulty of T tasks, abbreviated as the CPD-MTD formula. $\alpha$ denotes the global intercept term. $U_m$ represents the model - specific capability parameter, where $U_{m_0} = 0$ for the reference model. $V_t$ represents the task - specific difficulty parameter, where $V_{t_0} = 0$ for the reference task. The parameters can be solved using simple ordinary least squares as Formula 5.

Subsequently, the scores of each model on each benchmark are substituted into Formula 11, and the least squares method is used to fit the model capability coefficients $U_{mi}$ of the other models (except the reference model) and the task difficulty coefficients $V_{ti}$ of the other tasks (except the reference task), which ensures that the final comparison results are not affected by the reference task.

$$\text{TPP}_i = \exp\left(-V_{t_i} + \frac{1}{T}\sum_{j=1}^{T} V_{t_j}\right), \qquad \text{MPP}_j = \exp\left(U_{m_j} - \frac{1}{M}\sum_{k=1}^{M} U_{m_k}\right). \tag{12}$$

Finally, we take the exponential of the fitted $U_m$ and $V_m$ and perform geometric mean normalization to obtain the Model Power Parity (MPP) of each model and the Task Power Parity (TPP) of each benchmark, as shown in Formula 12. It is worth noting that we take the negative sign for $V_t$, because tasks with greater difficulty generally have lower scores.

## 2.4 PERFORMANCE PREDICTION

The CPD-MTD formula can also predict models' performance on some of the benchmarks without doing inference. We use the formula below to estimate the capability parameter of the model whose

Table 3: Top-3 masked benchmark combinations with the smallest prediction point error (MAE).

| Model | Best (MAE)↓ | 2ⁿᵈ best (MAE)↓ | 3ʳᵈ best (MAE)↓ |
|---|---|---|---|
| DeepSeek-R1-Distill-Qwen-1.5B | M-500, GSM8K (0.46) | M-500 (0.50) | GSM8K, GPQA (0.52) |
| DeepSeek-R1-Distill-Qwen-7B | M-500 (0.32) | M-500, GPQA (0.74) | M-500, GSM8K (0.93) |
| DeepSeek-R1-Distill-Llama-8B | HMMT (0.06) | HMMT, M-pro (0.30) | M-pro (0.58) |
| DeepSeek-R1-Distill-Qwen-14B | M-500 (1.28) | A-90, M-500, GSM8K (1.30) | A-90, M-500 (1.67) |
| DeepSeek-R1-Distill-Qwen-32B | M-500 (0.31) | M-500, HMMT (0.69) | A-90, M-500, HMMT (0.81) |
| QwQ-32B | M-500 (0.11) | M-500, HMMT (0.26) | HMMT (0.55) |
| DeepSeek-R1-Distill-Llama-70B | M-500 (0.88) | A-90, M-500 (1.20) | A-90, M-500, GSM8K, M-pro (1.33) |
| Qwen3-0.6B | GSM8K (1.60) | GSM8K, M-pro (2.72) | GSM8K, LiveCode (3.09) |
| Qwen3-1.7B | M-500 (1.00) | M-500, HMMT (1.60) | M-500, GSM8K, HMMT (1.67) |
| Qwen3-4B | M-500 (0.47) | M-500, GSM8K (0.77) | GSM8K (1.27) |
| Qwen3-8B | M-500 (0.01) | M-500, GSM8K (1.15) | A-90, M-500, GSM8K (1.43) |
| Qwen3-14B | M-500 (0.72) | A-90, M-500, GSM8K (0.92) | M-500, GSM8K (1.24) |
| Qwen3-32B | M-500 (0.35) | M-500, GSM8K, HMMT (1.38) | M-500, GSM8K (1.47) |
| Phi-4-mini-reasoning | M-500 (0.19) | M-500, M-pro (0.34) | M-500, GSM8K, M-pro (0.38) |
| Phi-4-reasoning | M-500 (1.30) | M-500, GSM8K, HMMT (1.56) | M-500, GSM8K, LiveCode, HMMT (1.94) |

Abbreviations: M-500 = MATH-500, A-90 = AIME-90, HMMT = HMMT-2025, M-pro = MMLU-Pro, GPQA = GPQA-Diamond, LiveCode = LiveCodeBench.

performance is to be predicted:

$$\hat{U}_{m^*} = \frac{1}{|K|} \sum_{t \in K} \left( y_{m^*,t} - \hat{\alpha} - \hat{V}_t \right) \quad (\text{if } K = \emptyset, \hat{U}_{m^*} = 0). \tag{13}$$

First, $\hat{\alpha}$ and $\{\hat{V}_t\}$ represent the estimated global intercept term and the set of estimated difficulty parameters for all tasks, respectively. These parameters can be directly derived through calculations using the previously introduced CPD-MTD formula. Parameter $y_{m^*,t}$ represents the observed score of model $m^*$ on task t. K denotes the set of tasks used to estimate the ability of model $m^*$. If K is an empty set ($K = \emptyset$), it means there are no tasks available for estimating the model's ability, and thus $\hat{U}_{m^*}$ is set to 0. Specifically, for the model $m^*$ to be evaluated, its estimated capability parameter $\hat{U}_{m^*}$ quantifies the inherent performance difference of $m^*$ relative to the baseline. In subsequent steps, $\hat{U}_{m^*}$ serves as the input in Formula 11 for estimating the performance of $m^*$ on unknown tasks.

## 2.5 PERPLEXITY AND REASONING ABILITY

We observe that the community selects different benchmarks when evaluating the ability of different types of models. For instance, distinct benchmarks are used for evaluating base models and reasoning models (Yang et al., 2025). We argue that when evaluating a specific type of model, benchmarks should be selected based on certain criteria; otherwise, the test results may fail to objectively reflect the model's ability. Perplexity (PPL) is commonly used to evaluate the basic text prediction ability of models. It is often one of the indicators for the loss function in the pre-training phase (Touvron et al., 2023) or applied to assess the ability of base models (Lin et al., 2024; Frantar et al., 2022; Dettmers et al., 2022), but it is rarely used for evaluating reasoning models. In this study, we take the PPL metric of reasoning models on WikiText-2 (Merity et al., 2016) as an example to analyze whether it is suitable for evaluating reasoning models.

Our method calculates the Spearman correlation (Hollander et al., 2013) between models' performance and capability parameter $U_m$, which is based on the assumption that the difficulty of a benchmark remains relatively stable for a specific category of models. For reasoning benchmarks, the capability parameter $U_m$ is fitted with the performance on the other reasoning benchmarks, then the Spearman correlation is calculated between the performance on the selected benchmark and $U_m$, which we call LOTO (Leave-One-Task-Out). The reason for this approach is that if all reasoning benchmarks are used for fitting, the information of the reasoning benchmarks intended for calculating the Spearman correlation may be incorporated into the fitted capability parameter $U_m$, thereby compromising the objectivity of the results. For perplexity benchmarks, the Spearman correlation is calculated between performance on them and $U_m$ fitted with performance on all reasoning benchmarks. The calculation is conducted on all available models.

## 3 EXPERIMENTS AND RESULTS

Based on the experimental setup proposed by (Liu et al., 2025), we evaluated over 30 reasoning models on 8 reasoning benchmarks. These models' parameter sizes span from 0.6B to 70B, and

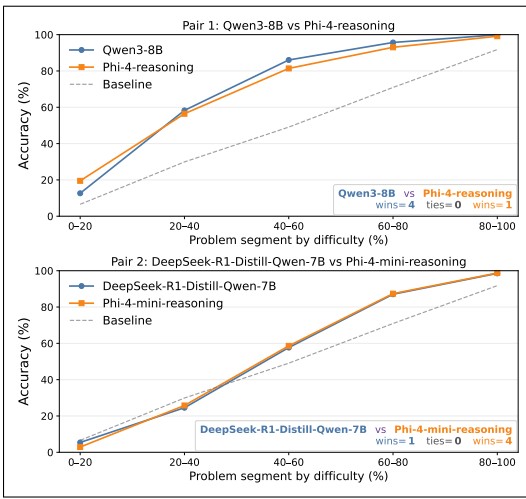

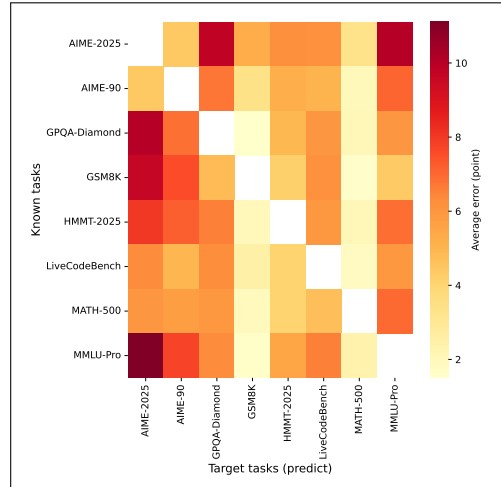

Figure 3: Accuracy comparison on different difficulty segments. The "wins" means the number of difficulty segments on which a model has higher accuracy.

Figure 4: Cross prediction error among reasoning benchmarks. The tasks with known performance are used to estimate the performance on target tasks.

quantized models (4-bit and 3-bit) are also included to build a larger model base. Mainstream open source reasoning large language models, including Qwen3 (Yang et al., 2025), DeepSeek-R1 (Guo et al., 2025), Phi-4 (Xu et al., 2025; Abdin et al., 2025), and QwQ (Qwen Team, 2025) are involved to reduce the parity on a specific model series. The benchmarks include high-level math problems (AIME, 2025; HMMT, 2025; AIME, 2024), math reasoning (Lightman et al., 2023; Cobbe et al., 2021), coding tasks (Jain et al., 2024), multitask understanding (Wang et al., 2024) and scientific problems (Rein et al., 2023). Our subsequent data analyses are based on every model's inference results on every benchmark, and the complete performance table is provided in Table 6 in the Appendix.

## 3.1 TPP AND MPP RANKINGS

We calculate the Model Power Parity (MPP) for 32 models and the Task Power Parity (TPP) for 8 benchmarks. The radar chart in Figure 2 presents the scores of the four reasoning models across various benchmarks, along with the corresponding TPP values for each benchmark. It can be readily observed that the difficulty ranking derived from TPP scores is consistent with that reflected by the average scores: HMMT Feb 25 exhibits the highest difficulty, while GSM8K and MATH-500 are the least difficult. For more details, Table 7 in the Appendix presents all the TPP scores of 8 benchmarks alongside the average scores of all models on each benchmark.

Table 1 presents the MPP results of 16 models selected from all 32 models. Results show that the Qwen3-32B model exhibits the strongest reasoning ability, while the Qwen3-0.6B model shows the weakest. Overall, for models within the same series, such as the Deepseek-R1-Distill series, Qwen3 series, and Phi-4 series, the reasoning ability of models consistently improves with increasing parameter scale. Across different model series, newer and more advanced models demonstrate better performance at the same parameter scale (Qwen3 outperforms Deepseek-R1-Distill). The complete MPP list is provided in Table 8 in the Appendix.

We compare the rankings derived by MPP and by average score. Only 8 models have changed their ranks, and all changes involve swapping positions with adjacent models. This indicates that the reasoning consistency of different models across various reasoning benchmarks is relatively high, and their capability improvements are balanced. Among the 8 models with ranking changes, we select two representative model pairs: Qwen3-8B vs. Phi-4-reasoning, and DeepSeek-R1-Distill-Qwen-7B vs. Phi-4-mini-reasoning. According to the ranking based on average scores, Phi-4-reasoning and DeepSeek-R1-Distill-Qwen-7B perform better; however, the opposite is true when ranked by MPP. We categorize each question from all 8 benchmarks into five difficulty groups based on the average accuracy of responses from all models, and count the number of correct answers each

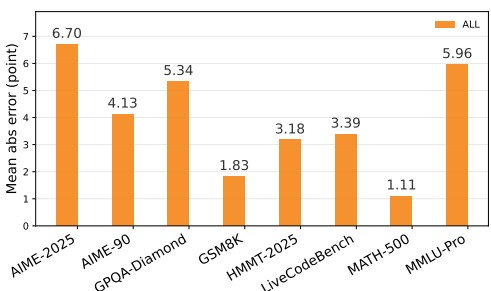

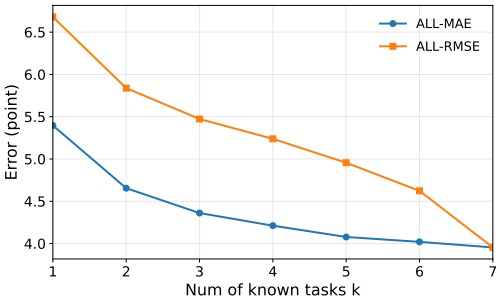

Figure 5: MAE of performance prediction when the number of known tasks $k = 7$. "All" means the performance of all models is considered.

Figure 6: The MAE and RMSE curve between performance prediction point error and the number of known tasks $k$.

model achieves within each difficulty segment, which is shown in Figure 7 in the Appendix. Among them, the bar chart in Figure 2 depicts the proportions of questions at different difficulty levels that the aforementioned four models answer correctly or incorrectly across each benchmark. To make the presentation more intuitive, Figure 3 plots the performance of the two model pairs across different difficulty groups. It can be observed that Qwen3-8B and Phi-4-mini-reasoning outperform Phi-4-reasoning and DeepSeek-R1-Distill-Qwen-7B in more difficult segments, respectively, and the baseline represents the average accuracy over all models. **This suggests that the MPP ranking focuses on the overall ability of models across different difficulty levels**, while assigning lower evaluations to models that only excel in easy or difficult questions, which aligns better with human judgment of capability.

## 3.2 BENCHMARK PERFORMANCE PREDICTION

We conduct prediction experiments under various settings. Specifically, we iterate through each model sequentially, adjusting the number $k$ of tasks on which the model's performance is known to predict the performance of the remaining $8 - k$ tasks. Table 2 presents the benchmark prediction results for 15 models under the condition of $k = 6$, where the two benchmarks with the smallest Mean Absolute Error (MAE) and Root Mean Square Error(RMSE) are selected for each model. It can be observed that, except for the Qwen3-0.6B model, which exhibits a relatively large prediction error (3.45 points) on MMLU-Pro, all other errors are within 2.8 points, and approximately 43% of the prediction results have an error within 0.5 points. This result can be considered relatively accurate. When reasoning large language models conduct inference on reasoning benchmarks, there are typically fluctuations of several points between different inference runs, and this observation indicates that the previous estimation of model capability parameters is relatively reliable.

Furthermore, we test the cross-prediction performance between benchmarks under the extreme scenario of $k = 1$, as shown in Figure 4. We observe that Figure 4 is roughly symmetric about the diagonal, which implies that the mutual prediction relationship between two benchmarks is similar. MMLU-Pro and GPQA-Diamond, respectively, exhibit large mutual prediction errors with AIME 25 and AIME-90, but their mutual prediction errors are relatively small. This indicates that the reasoning performance patterns of models on multi-domain knowledge question answering tasks differ to some extent from those on high difficulty mathematics competition problems. MATH-500 shows the smallest mutual prediction errors with other benchmarks, suggesting that performance on MATH-500 is highly consistent across models, and MATH-500 shares similar reasoning ability demand with other benchmarks. The value of $k$ can be changed to further explore the prediction relationships between different benchmark combinations.

When $k = 7$, we evaluate the average prediction error of each benchmark across all models, as illustrated in Figure 5. Among these, MATH-500 and GSM8K exhibit the smallest average error, which aligns with the observation, in Table 3 where these two benchmarks appear most frequently in the combinations with the smallest errors. Furthermore, the prediction errors do not show a significant positive correlation with the TPP values of the respective benchmarks. For instance, the prediction error of HMMT-2025 is smaller than that of MMLU-Pro. We attribute this to the fact that prediction errors are also influenced by the performance variations of every single model

Table 4: Five reasoning models and their base models' perplexity (PPL) on wikitext. The sequence length is set to 2048.

| Model | PPL↓ |
|---|---|
| Deepseek-R1-Distill-Qwen-1.5B | 40.80 |
| Qwen2.5-Math-1.5B | 17.68 |
| Deepseek-R1-Distill-Qwen-7B | 25.13 |
| Qwen2.5-Math-7B | 11.58 |
| Deepseek-R1-Distill-Qwen-14B | 8.92 |
| Qwen2.5-14B | 5.29 |
| Qwen3-1.7B | 16.72 |
| Qwen3-1.7B-base | 9.40 |
| Qwen3-4B | 13.65 |
| Qwen3-4B-base | 7.90 |

Table 5: Spearman $\rho$ between performance on each benchmark and model effect $U_m$. A larger value means a stronger correlation.

| Benchmark | Spearman ($\rho$)↑ |
|---|---|
| AIME-90 | 0.98 |
| MATH-500 | 0.97 |
| GSM8K | 0.97 |
| LiveCodeBench | 0.96 |
| AIME 25 | 0.96 |
| GPQA-Diamond | 0.95 |
| MMLU-Pro | 0.92 |
| HMMT Feb 25 | 0.90 |
| **Reasoning Bench Avg** | **0.95** |
| Wikitext | 0.74 |

on each benchmark, as well as the consistency of performance across different models on each benchmark. Models whose actual performance deviates significantly from the average performance of most models typically introduce larger prediction errors. This is because it is difficult to fully fit their capability parameter during the parameter estimation process.

In addition, without restricting the value of $k$, we test the results of all known benchmark combinations for $k$ ranging from 1 to 7. We then rank the average errors of all prediction results while imposing a constraint that the absolute difference between the predicted and actual results for any single benchmark must be less than 4 points. Table 3 presents the top three masked benchmark combinations that yield the smallest error for each model. In general, a larger k corresponds to lower MAE, which is also revealed in Figure 6. Furthermore, we observe that increasing $k$ from 5 to 7 does not lead to a notable decrease in prediction error. The key reason is that the estimation gradually stabilizes as $k$ grows: a larger $k$ introduces more input information, which reduces random fluctuations in the estimation output.

## 3.3 CHOOSING SUITABLE BENCHMARKS FOR EVALUATION

The Perplexity (PPL) results of several reasoning models and their corresponding base models are shown in Table 4. The PPL results of all base models outperform those of their corresponding reasoning models, which preliminarily indicates that there is no significant positive correlation between the PPL metric and reasoning ability.

Furthermore, we follow the analysis method in Section 2.5 and calculate the Spearman correlation coefficient between performance on each benchmark and model effect $U_m$. Table 5 indicates that the model's performance on reasoning benchmarks has a stronger correlation with its reasoning ability, while the PPL metric may mislead the judgment for reasoning model evaluation.

In addition, it is important to emphasize that while our analysis here takes WikiText-2 as an example, the proposed method is essentially a general approach. It is grounded on the assumption that the difficulty of a benchmark remains relatively stable for a specific category of models, and thus can also be applied to evaluate whether other benchmarks are suitable for reasoning models or other types of models.

# 4 LIMITATIONS AND FUTURE WORK

The MPP framework manages to predict models' performance without doing inference, but we find that the prediction error of some models on some benchmarks is relatively higher. This may be attributed to the unique characteristics of the models' inherent capabilities and the limitations of the mathematical models employed in our current evaluation framework. In the future, we will further expand the scale of evaluation, covering more benchmarks and models, and consider introducing question-level difficulty labels to enhance the precision of our existing mathematical models, thereby achieving more robust predictions. Additionally, we will explore decomposing model ca-

pabilities into domain-specific subfields, such as mathematical reasoning and code generation, and conduct fine-grained comparisons across specific benchmark-oriented tracks, aiming to evaluate more scenario-specific applications of models with distinct strengths.

## 5 RELATED WORK

### 5.1 BENCHMARK DIFFICULTY EVALUATION

GPQA (Rein et al., 2023) consists of graduate-level multiple-choice questions, whose difficulty is defined as the mean of two expert ratings on a 4-point scale. MedConceptsQA (Shoham & Rappoport, 2024) is a benchmark for medical concept QA, measuring difficulty using graph-based distances: in its undirected vocabulary graph, questions are harder when answer options are closer. LingOly (Bean et al., 2024), a linguistic reasoning benchmark, assigns questions to five levels based on semantic similarity to English and reasoning complexity. ConsisEval (Yang et al., 2024) encodes difficulty via strictly ordered question pairs, each comprising an easy problem from existing datasets and a harder counterpart generated either manually or automatically. Easy2Hard-Bench (Ding et al., 2024) aggregates six cross-domain datasets with standardized difficulty labels, offering new perspectives on benchmark difficulty estimation.

### 5.2 REASONING MODELS AND BENCHMARKS

Recent advances in reasoning models reflect a shift from general-purpose LLMs to models optimized for logical deduction, multi-step reasoning, and contextual understanding. OpenAI's o1(OpenAI, 2024) series introduced the notion of allocating additional computation to "thinking time", yielding substantial improvements in tasks such as mathematics and code reasoning. DeepSeek-R1(Guo et al., 2025) enhances reasoning capabilities through a combination of reinforcement learning and efficient distillation, demonstrating competitive performance at considerably lower training cost. Microsoft's Phi-4(Xu et al., 2025; Abdin et al., 2025) focused on lightweight deployment and reliable mathematical reasoning. Qwen3(Yang et al., 2025) series integrated both dense and Mixture-of-Expert (MoE) architectures to support stronger multi-step reasoning, extended context lengths, and multilingual applications.

To evaluate the models' reasoning ability, various reasoning benchmarks have been established in recent years. AIME-2025 (AIME, 2025), AIME-90 (AIME, 2024), and HMMT-2025 (HMMT, 2025) represent high-level mathematical competitions, requiring problem-solving strategies comparable to those of human contestants. GSM8K (Cobbe et al., 2021) and MATH-500 (Lightman et al., 2023) focus on multi-step arithmetic and proof-like reasoning, testing the ability to perform structured mathematical derivations. LiveCodeBench (Jain et al., 2024) evaluates program synthesis and execution in realistic coding scenarios. MMLU-Pro (Wang et al., 2024) covers a broad set of academic and professional domains, providing a comprehensive measure of multi-domain understanding. GPQA (Rein et al., 2023) assesses advanced scientific reasoning through graduate-level questions that require domain-specific knowledge and conceptual inference.

## 6 CONCLUSION

In this paper, we introduce MPP, a simple and novel multilateral comparison framework for benchmark difficulty estimation and reasoning large language model ranking. Extensive experiments validate the comparability between the two pairs of concepts: currencies and goods, as well as LLMs and benchmarks. We conduct evaluations on over 30 reasoning models and 8 prevalent reasoning benchmarks, revealing that the MPP framework can directly predict the remaining performance results based on partial models' performance. Furthermore, by analyzing the magnitude of prediction errors, we demonstrate the derived correlations among different reasoning benchmarks, which reflect the intrinsic connections of reasoning models across various domains of reasoning tasks. Moreover, we argue that benchmarks should be selectively chosen based on the specific type of models. Results indicate that for reasoning LLMs, the perplexity metric is not suitable for measuring the strength of their reasoning capabilities. To sum up, we hope that MPP will provide the community with a simple yet novel framework for model ability comparison and benchmark difficulty assessment, facilitating the construction of benchmarks that more accurately reflect the true capabilities of models and further enhancing the granularity and professionalism of LLM evaluations.

REPRODUCIBILITY STATEMENT

The code used in this paper can be found in the link here. Clone the repo, open it in any code editor, and run the scripts in a Python environment. You can use the data provided in the paper or use your own results. The benchmarks used in this paper are all publicly available online.

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

# APPENDIX

# A THE USE OF LARGE LANGUAGE MODELS (LLMS)

During the manuscript writing process, we utilize Large Language Models (LLMs) to assist with polishing certain linguistic expressions and checking for grammatical errors. Additionally, we leverage LLMs to search and review relevant literature and materials.

# B PURCHASING POWER PARITY CALCULATION THEORY

The PPP theory below mainly refers to the release of World Bank (Rao, 2013).

## B.1 JEVSON-GEKS

The Jevons index calculates Purchasing Power Parity (PPP) by taking the geometric mean of the price ratios for all goods between two countries. Assume that all items are priced in all countries with no weights assigned to their representativeness or importance. In the simplest case where all items are priced in all countries and weighted equally, the PPP at a given basic heading level can be computed using:

$$PPP_{jk}^{Jevons} = \prod_{i=1}^{N} \left[ \frac{p_i^k}{p_i^j} \right]^{\frac{1}{N}}. \quad \text{for all j,k = 1,2...,C.} \tag{14}$$

## B.2 THE SOLUTION FOR CPD FORMULA

Now consider the case in which all items in the basic heading are priced in all countries. In this case, for the aggregation at the basic heading level where there are no weights, the parameters $\alpha_j$ and $\eta_i$ can be estimated using simple unweighted or ordinary least squares by minimizing:

$$\sum_{i=1}^{N} \sum_{j=1}^{C} (\ln p_{ij} - \alpha_j - \gamma_i)^2. \tag{15}$$

The first-order conditions for optimization with respect to $\alpha_j$ and $\eta_i$ lead to the following system of $C + N$ equations in as many unknowns:

$$\alpha_j = \frac{1}{N} \sum_{i=1}^{N} \ln p_{ic} - \frac{1}{N} \sum_{i=1}^{N} \gamma_i, \text{ for } j = 1, 2, \ldots, C \text{ and} \tag{16}$$

$$\gamma_i = \frac{1}{C} \sum_{j=1}^{C} \ln p_{ic} - \frac{1}{C} \sum_{j=1}^{C} \alpha_c, \text{ for } n = 1, 2, \ldots, N. \tag{17}$$

This system can be solved by imposing a linear restriction on the unknown parameters. For example, if $\alpha_1 = 0$ is the restriction imposed, it can be easily shown that, for each $j = 2, \ldots, C$:

$$\hat{\alpha}_j = \frac{1}{N} \sum_{i=1}^{N} [\ln p_{nj} - \ln p_{n1}] \quad \text{or} \quad PPP_j = \exp(\hat{\alpha}_j) = \left[ \prod_{i=1}^{N} \frac{p_{ij}}{p_{i1}} \right]^{\frac{1}{N}}. \tag{18}$$

Using the solution in (4.20), comparisons of price levels between countries $j$ and $k$, represented by $PPP_{jk}$, can be derived as below, which is obviously transitive and base-invariant:

$$PPP_{jk} = \frac{\exp(\hat{\alpha}_k)}{\exp(\hat{\alpha}_j)} = \left[ \prod_{i=1}^{N} \frac{p_{ik}}{p_{ij}} \right]^{\frac{1}{N}}. \tag{19}$$

### B.3 JEVSON-GEKS-MTD FORMULA

Furthermore, we take the natural logarithm of both sides of Equation 14 and reconstruct the multi-lateral comparison relationships between different models and benchmarks, which is referred to as the JEVSON-GEKS-MTD formula:

$$\ln F_{a,b} = \frac{1}{|M_{a,b}|} \sum_{m \in M_{a,b}} (y_{m,a} - y_{m,b}), \qquad \ln \text{TPP}_{\text{raw}}(t) = \frac{1}{|T|} \sum_{s \in T} \left(\ln F_{t,s} - \ln F_{\text{ref},s}\right),$$

$$\text{TPP}(t) = \exp\!\left(-\ln \text{TPP}_{\text{raw}}(t) + \frac{1}{|T|} \sum_{u \in T} \ln \text{TPP}_{\text{raw}}(u)\right) \tag{20}$$

$$\ln F_{a,b} = \frac{1}{|T_{a,b}|} \sum_{t \in T_{a,b}} (y_{a,t} - y_{b,t}), \qquad \ln \text{MPP}_{\text{raw}}(m) = \frac{1}{|M|} \sum_{u \in M} \left(\ln F_{m,u} - \ln F_{\text{ref},u}\right),$$

$$\text{MPP}(m) = \exp\!\left(\ln \text{MPP}_{\text{raw}}(m) - \frac{1}{|M|} \sum_{v \in M} \ln \text{MPP}_{\text{raw}}(v)\right) \tag{21}$$

Let $M$ denote the set of models and $T$ denote the set of tasks. $y_{m,t}$ denote the score of model $m$ on task $t$. $M_{a,b}$ represents the set of models that have scores on both task $a$ and task $b$. $T_{a,b}$ represents the set of tasks for which there are scores of on both model $a$ and model $b$. And ref is an arbitrarily chosen reference task or model.

## C INFERENCE PERFORMANCE

Table 6: All 32 models' performance on all 8 reasoning benchmarks. The suffix "awq" or "gptq" refers to the quantization methods, and "w3" or "w4" means weight-only 3-bit or 4-bit precision. The full score is 100 for all benchmarks.

| Model Name | AIME-90 | AIME-2025 | MATH-500 | GSM8K | GPQA-Diamond | LiveCodeBench | HMMT-2025 | MMLU-Pro |
|---|---|---|---|---|---|---|---|---|
| DeepSeek-R1-Distill-Qwen-1.5B | 22.22 | 20.00 | 85.60 | 83.78 | 34.85 | 14.93 | 10.00 | 32.86 |
| DeepSeek-R1-Distill-Qwen-1.5B-awq-w4 | 25.56 | 26.67 | 83.60 | 83.24 | 27.78 | 13.81 | 13.33 | 31.43 |
| DeepSeek-R1-Distill-Qwen-1.5B-gptq-w4 | 22.22 | 23.33 | 82.80 | 83.47 | 34.85 | 15.67 | 6.67 | 35.71 |
| DeepSeek-R1-Distill-Qwen-1.5B-awq-w3 | 5.56 | 0.00 | 45.00 | 63.38 | 24.75 | 3.73 | 0.00 | 24.29 |
| DeepSeek-R1-Distill-Qwen-1.5B-gptq-w3 | 12.22 | 16.67 | 67.80 | 72.25 | 25.76 | 7.09 | 3.33 | 24.29 |
| DeepSeek-R1-Distill-Qwen-7B | 47.78 | 36.67 | 93.40 | 91.13 | 54.04 | 38.06 | 13.33 | 55.71 |
| DeepSeek-R1-Distill-Qwen-7B-awq-w4 | 41.11 | 36.67 | 92.80 | 90.52 | 47.98 | 39.55 | 13.33 | 57.14 |
| DeepSeek-R1-Distill-Qwen-7B-gptq-w4 | 47.78 | 33.33 | 93.60 | 91.74 | 47.98 | 32.09 | 16.67 | 45.71 |
| DeepSeek-R1-Distill-Qwen-7B-awq-w3 | 30.00 | 33.33 | 89.20 | 89.92 | 44.95 | 25.75 | 10.00 | 50.00 |
| DeepSeek-R1-Distill-Qwen-7B-gptq-w3 | 28.89 | 23.33 | 90.40 | 89.69 | 38.38 | 26.12 | 10.00 | 54.29 |
| DeepSeek-R1-Distill-Llama-8B | 42.22 | 26.67 | 88.40 | 88.63 | 40.91 | 36.57 | 13.33 | 58.57 |
| DeepSeek-R1-Distill-Llama-8B-awq-w4 | 31.11 | 23.33 | 88.00 | 87.04 | 48.48 | 34.70 | 13.33 | 57.14 |
| DeepSeek-R1-Distill-Llama-8B-gptq-w4 | 37.78 | 33.33 | 90.20 | 88.48 | 46.97 | 34.33 | 16.67 | 55.71 |
| DeepSeek-R1-Distill-Llama-8B-awq-w3 | 16.67 | 30.00 | 78.80 | 83.24 | 34.85 | 23.13 | 10.00 | 47.14 |
| DeepSeek-R1-Distill-Llama-8B-gptq-w3 | 22.22 | 26.67 | 83.00 | 85.52 | 34.85 | 26.12 | 20.00 | 44.29 |
| DeepSeek-R1-Distill-Qwen-14B | 62.22 | 53.33 | 94.80 | 94.01 | 58.08 | 51.49 | 30.00 | 71.43 |
| DeepSeek-R1-Distill-Qwen-32B | 63.33 | 56.67 | 96.60 | 94.24 | 64.14 | 57.84 | 26.67 | 70.00 |
| QwQ-32B | 80.00 | 73.33 | 97.40 | 95.45 | 63.13 | 60.82 | 36.67 | 78.57 |
| DeepSeek-R1-Distill-Llama-70B | 64.44 | 53.33 | 95.60 | 94.01 | 65.15 | 55.97 | 23.33 | 81.43 |
| Qwen3-0.6B | 12.22 | 3.33 | 72.60 | 78.32 | 29.29 | 13.81 | 13.33 | 35.71 |
| Qwen3-0.6B-awq-w4 | 8.89 | 3.33 | 62.20 | 68.31 | 26.26 | 9.33 | 6.67 | 24.29 |
| Qwen3-1.7B | 38.89 | 33.33 | 91.20 | 89.76 | 37.37 | 31.72 | 16.67 | 65.71 |
| Qwen3-1.7B-awq-w4 | 32.22 | 20.00 | 89.20 | 88.93 | 34.34 | 22.76 | 10.00 | 55.71 |
| Qwen3-1.7B-awq-w3 | 1.11 | 0.00 | 44.20 | 65.96 | 29.80 | 0.00 | 3.33 | 35.71 |
| Qwen3-4B | 58.89 | 63.33 | 96.60 | 94.84 | 53.03 | 54.10 | 30.00 | 68.57 |
| Qwen3-4B-awq-w4 | 61.11 | 46.67 | 95.80 | 94.31 | 52.02 | 48.13 | 23.33 | 70.00 |
| Qwen3-4B-awq-w3 | 25.56 | 30.00 | 87.40 | 89.61 | 35.35 | 21.64 | 13.33 | 60.00 |
| Qwen3-8B | 68.89 | 63.33 | 96.80 | 94.77 | 62.12 | 56.72 | 36.67 | 71.43 |
| Qwen3-14B | 71.11 | 70.00 | 96.60 | 95.75 | 64.14 | 62.69 | 40.00 | 74.29 |
| Qwen3-32B | 80.00 | 70.00 | 97.20 | 95.30 | 69.19 | 61.94 | 40.00 | 75.71 |
| Phi-4-mini-reasoning | 47.78 | 33.33 | 93.60 | 93.63 | 47.47 | 27.24 | 20.00 | 65.71 |
| Phi-4-reasoning | 72.22 | 66.67 | 95.60 | 94.01 | 64.65 | 55.60 | 33.33 | 72.86 |

## D TPP AND MPP FULL LIST

It should be noted that the TPP values of benchmarks and the MPP values of models are compared at the "odds" level (see Section 3.1). While these values can be used for ability ranking comparisons, they do not directly imply that a specific model is "equivalent" to another model in a quantitative sense.

Table 7: Task Power Parity (TPP) and average score on reasoning benchmarks.

| Metric Type | HMMT Feb 25 | AIME 25 | LivecodeBench | AIME-90 | GPQA-Diamond | MMLU-Pro | GSM8K | MATH-500 |
|---|---|---|---|---|---|---|---|---|
| TPP (Difficulty) | 6.55 | 3.42 | 2.84 | 1.63 | 1.13 | 0.75 | 0.11 | 0.10 |
| Average Score (%) | 17.92 | 35.31 | 33.23 | 40.07 | 45.09 | 54.73 | 87.29 | 86.12 |

Table 8: Full MPP ranking for all reasoning models. The results are retained to four decimal places.

| Model | MPP↑ |
|---|---|
| Qwen3-32B | 4.2119 |
| QwQ-32B | 4.1999 |
| Qwen3-14B | 3.7864 |
| Qwen3-8B | 3.2660 |
| Phi-4-reasoning | 3.1936 |
| DeepSeek-R1-Distill-Llama-70B | 2.8575 |
| DeepSeek-R1-Distill-Qwen-32B | 2.8490 |
| Qwen3-4B | 2.7415 |
| DeepSeek-R1-Distill-Qwen-14B | 2.5311 |
| Qwen3-4B-awq-w4 | 2.2827 |
| Phi-4-mini-reasoning | 1.5400 |
| DeepSeek-R1-Distill-Qwen-7B | 1.4663 |
| DeepSeek-R1-Distill-Qwen-7B-awq-w4 | 1.3673 |
| DeepSeek-R1-Distill-Qwen-7B-gptq-w4 | 1.3468 |
| Qwen3-1.7B | 1.2535 |
| DeepSeek-R1-Distill-Llama-8B-gptq-w4 | 1.2217 |
| DeepSeek-R1-Distill-Llama-8B | 1.1332 |
| DeepSeek-R1-Distill-Llama-8B-awq-w4 | 1.0408 |
| DeepSeek-R1-Distill-Qwen-7B-awq-w3 | 0.9982 |
| Qwen3-4B-awq-w3 | 0.9399 |
| DeepSeek-R1-Distill-Qwen-7B-gptq-w3 | 0.9350 |
| Qwen3-1.7B-awq-w4 | 0.8729 |
| DeepSeek-R1-Distill-Llama-8B-gptq-w3 | 0.8292 |
| DeepSeek-R1-Distill-Llama-8B-awq-w3 | 0.6878 |
| DeepSeek-R1-Distill-Qwen-1.5B-awq-w4 | 0.6375 |
| DeepSeek-R1-Distill-Qwen-1.5B | 0.6214 |
| DeepSeek-R1-Distill-Qwen-1.5B-gptq-w4 | 0.5989 |
| Qwen3-0.6B | 0.3883 |
| DeepSeek-R1-Distill-Qwen-1.5B-gptq-w3 | 0.3115 |
| Qwen3-0.6B-awq-w4 | 0.2584 |
| DeepSeek-R1-Distill-Qwen-1.5B-awq-w3 | 0.0226 |
| Qwen3-1.7B-awq-w3 | 0.0202 |

# E QUANTIZATION

To enhance the efficiency of large language models (LLMs), *weight-only quantization* is commonly employed. This technique reduces both model size and computational cost by representing full-precision weights $W$ with low-bit quantized weights $\hat{W}$, typically using a linear quantization-dequantization scheme:

$$\hat{\mathbf{X}} = \mathcal{Q}(\mathbf{X}; b) = s \cdot \Pi_{\Omega(b)}\left(\frac{\mathbf{X}}{s}\right), \tag{22}$$

where $s = \frac{\max(\mathbf{X}) - \min(\mathbf{X})}{2^b - 1} \in \mathbb{R}^+$ and $\Pi(\cdot)$ denotes the projection onto the nearest element in the set $\Omega(b)$ of $b$-bit integers. Among weight-only approaches, **Gradient Post-Training Quantization (GPTQ)** (Frantar et al., 2022) and **Activation-aware Weight Quantization (AWQ)** (Lin et al., 2024) are the most widely used. GPTQ exploits second-order information to minimize accuracy loss under low-bit settings, while AWQ further enhances robustness by preserving weights that are

highly sensitive to activation variance. In our experiments, we include models quantized with both GPTQ and AWQ methods.

## F  PROBLEM-LEVEL PERFORMANCE

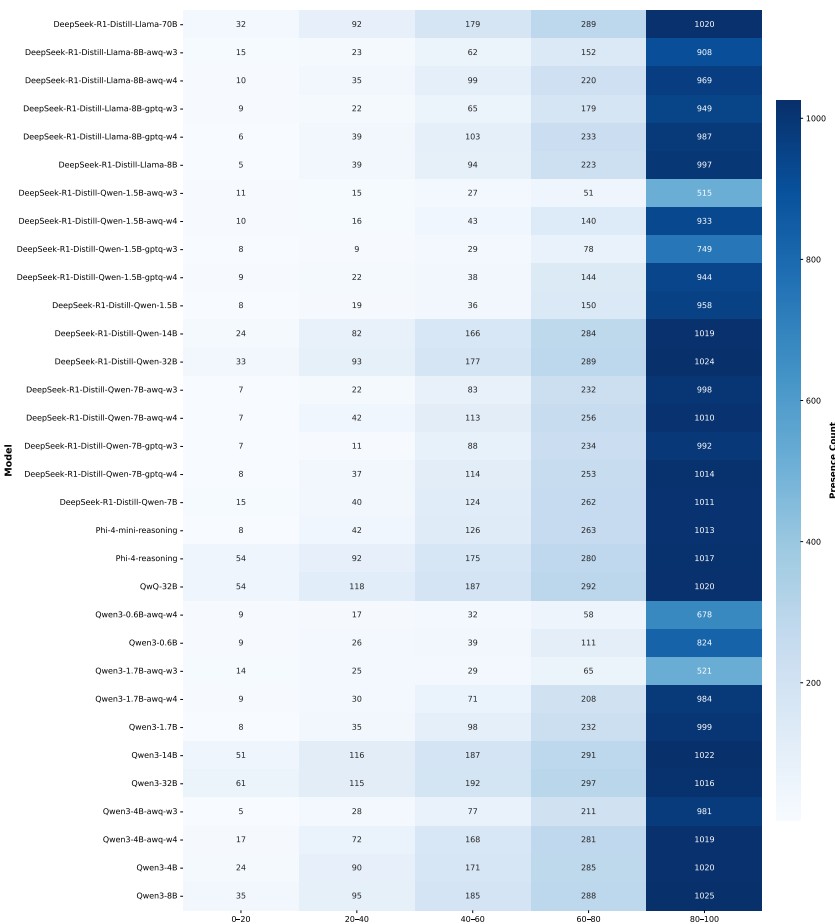

Figure 7: All reasoning models' performance on problems across different difficult levels.

