# OpenReview forum: "MPP: MODEL POWER PARITY FOR REASONING LARGE LANGUAGE MODEL COMPARISON ON CROSS-DOMAIN BENCHMARKS"
_ICLR.cc/2026/Conference — Submitted to ICLR 2026_

### Official Review · Reviewer_EHvj · 2025-10-20

**Soundness:** 2
**Presentation:** 1
**Contribution:** 2
**Rating:** 2
**Confidence:** 3

**Summary:**

This paper offers a new method inspired by economics to evaluate language model abilities and benchmark suitability. It adapts Purchasing Power Parity (PPP) and the Country Product Dummy (CPD) method to essentially infer a normalized or equalized version of the raw aggregate score on a benchmark. These methods are used to evaluate many models on several benchmarks. The ranking of models is preserved, the raw score can be recovered from the PPP, and the suitability of a benchmark for evaluating a new model can be determined.

**Strengths:**

The idea behind this paper is novel. The authors clearly expended considerable effort evaluating and analysing data from some 30 models on 8 reasoning tasks. The fact that raw score can be recovered from the inferred PPP is a useful result, but not unexpected given the simplicity of the regression models.

**Weaknesses:**

It is not clear how the PPP is computed with the CPD method. As far as I can tell, CPD is a regression with binary dummy input variables encoding country and product, with the aim to predict the price of the product in that country. But of course, each price is perfectly identifiable just from the matrix of dummy variables. The authors do not describe how to overcome this identification problem. As far as I can tell (not being an economist myself), there are some methods for handling this (omitting the reference categories, etc.), but the authors do not describe these methods in detail. This is essential for their method to be taken up by the community. Indeed, throughout, the methodology is described only at a very high-level, making implementation based on the text almost impossible without independent knowledge of these methods. The authors should more clearly describe their methodology step-by-step.

While the idea is innovative, I do wonder what the added utility of computing the MPP is over just reporting the per-benchmark aggregate scores. Both methods gloss over the crucial instance-level variation contained in these benchmarks, which is far more informative with respect to the capabilities of the model than aggregating (Burnell et al., 2023; Raji et al., 2021). The MPP method just adds another, albeit slightly more sophisticated, way to compute the average performance on large datasets. AI Evaluation has moved on these days to the examination of the individual demands of specific items and measuring how that impacts performance, which would give us a much better sense of what models can and cannot do. In that vein, the authors should review and compare to the literature on Item Response Theory and psychometrics for AI evaluation: Polo et al. (2024), Jo and Wilson (2025), Burden et al. (2023), Kipnis et al. (2024), Wang et al. (2023).

In general, the paper is vague and unclear. The notation is quite inconsistent throughout. For instance, PPP is variously indexed by two countries/models or by one (it should properly be with two always), the formatting in equations (4) and (7) is poor, and the authors do not define or justify certain steps, such as how they normalize raw scores or why scores are log-transformed (is it for numerical stability or monotonicity?). I also dislike that the important definitions relating to PPP and CPD are all included in the appendix, demanding the reader to flick back and forth to gain an understanding of the methods and results.


Burden, J., Voudouris, K., Burnell, R., Rutar, D., Cheke, L., & Hernández-Orallo, J. (2023). Inferring capabilities from task performance with bayesian triangulation. arXiv preprint arXiv:2309.11975.

Burnell, R., Schellaert, W., Burden, J., Ullman, T. D., Martinez-Plumed, F., Tenenbaum, J. B., ... & Hernandez-Orallo, J. (2023). Rethink reporting of evaluation results in AI. Science, 380(6641), 136-138.

Jo, N., & Wilson, A. (2025). What Does Your Benchmark Really Measure? A Framework for Robust Inference of AI Capabilities. arXiv preprint arXiv:2509.19590.

Kipnis, A., Voudouris, K., Buschoff, L. M. S., & Schulz, E. (2024). metabench--A Sparse Benchmark of Reasoning and Knowledge in Large Language Models. arXiv preprint arXiv:2407.12844.

Polo, F. M., Weber, L., Choshen, L., Sun, Y., Xu, G., & Yurochkin, M. (2024). tinyBenchmarks: evaluating LLMs with fewer examples. arXiv preprint arXiv:2402.14992.

Raji, I. D., Bender, E. M., Paullada, A., Denton, E., & Hanna, A. (2021). AI and the everything in the whole wide world benchmark. arXiv preprint arXiv:2111.15366.

Wang, X., Jiang, L., Hernandez-Orallo, J., Stillwell, D., Sun, L., Luo, F., & Xie, X. (2023). Evaluating general-purpose AI with psychometrics. arXiv preprint arXiv:2310.16379.

**Questions:**

1. It seems that one big component of this paper is to simply see if we can recover the ranking and raw scores from the MPP. Why should a practitioner bother computing the MPP at all, if all we end up caring about is the (ranking of the) raw scores on the benchmark?
2. What method is used to avoid collinearity in the CPD regression?
3. How does the TPP method compare to more sophisticated methods like Item Response Theory, which allow the computation of difficulty on a per-item, rather than per-benchmark, basis?
4. Is recovery of raw scores within 3 percentage points good? Kipnis et al. (2025) appear to achieve recover with much lower errors (0.5-1.24%)
5. The title and much of the discussion focuses on reasoning models, but this method seems agnostic to whether they are reasoning models or not. Indeed, you could use it to compare deep RL agents or computer vision systems, mutatis mutandi. Why have the authors focused on reasoning LLMs and reasoning benchmarks?

---

> ### Author Response · Authors · 2025-11-21
> **Rebuttal by Authors**
>
> Thank you for your thoughtful and constructive comments.  Below we provide point-by-point responses:
>
> W1: Lack of explanation of the Country Product Dummy (CPD) method.
>
> W1-Ans: We provide a step-by-step explanation of the procedure in the **revised version** (Section 2.2).
>
> W2: The utility of MPP and comparison with Item Response Theory.
>
> W2-Ans: Please refer to Q1-Ans, Q3-Ans and Q4-Ans.
>
> W3: Issues in writing clarity, methodological transparency and structural organization.
>
> W3-Ans:  We revise the formatting and definitions of equations to improve precision and readability.
>
> Q1: Why computing  MPP?
>
> Q1-Ans: We first compute the MPP values for all models and verify the MPP framework yields a model capability rankings that aligns with general intuition as shown in Table 1. We address your question by considering two scenarios:
>
> (1) Complete model-benchmark scores
>
> In this case, MPP is used to place model capabilities on a comparable scale, enabling **aggregated comparisons across heterogeneous benchmarks** as illustrated in **Figure 1**.
>
> (2) Incomplete model-benchmark scores or prediction
>
> The computation of MPP can operate under partially missing benchmark scores as shown in **Table 2**.
> This property helps save time and computational resources. For example, given a maintained leaderboard containing the performance of existing models on major benchmarks, one may only need to evaluate a newly released model on **a subset of benchmarks**.
>
> Q2: What method is used to avoid collinearity in the CPD regression?
>
> Q2-Ans:  The CPD method imposes the normalization constraint that the reference country has PPP=1, and then estimates the PPP  of the remaining countries relative to this reference.
>
> Q3:  Comparison to Item Response Theory?
>
> Q3-Ans:  We view the MPP framework and Item Response Theory (IRT)  as **complementary tools** rather than competing methodologies. Their distinctions can be summarized as follows:
>
> (1) Different purposes
>
> IRT is designed to explain **item-level response patterns**, with the goal of modeling:
>
> - a difficulty parameter for **each individual item (question)**,
> - an ability parameter for **each respondent (model)**,
>
> MPP framework:
>
> - **normalizing heterogeneous benchmarks onto a comparable scale**,
> - **characterizing the overall difficulty of an entire benchmark**,
>
> (2) Data source
>
> IRT requires a relatively rich response matrix:
>
> - dozens to hundreds of models to stabilize item difficulty estimates,
> - typically tens of thousands of items to reliably estimate abilities,
> - but item-level correctness is often **not released** (Burnell et al. 2023),
>
> **Kipnis et al. (2024) analyses over 5000 LLMs.**
>
> However, in our evaluation setting:
>
> - some benchmarks (e.g., AIME 25) contain only 30 items,
> - the number of models is limited (32),
> - and responses for items are highly imbalanced (most models all-correct or all-wrong).
>
> Under this setting, item-level IRT produces:
>
> - highly unstable difficulty estimates,
> - and convergence difficulties even under 1PL or 2PL models.
>
> Burnell, R., Schellaert, W., Burden, J., Ullman, T. D., Martinez-Plumed, F., Tenenbaum, J. B., ... & Hernandez-Orallo, J. (2023). Rethink reporting of evaluation results in AI. Science, 380(6641), 136-138.
>
> Kipnis, A., Voudouris, K., Buschoff, L. M. S., & Schulz, E. (2024). metabench--A Sparse Benchmark of Reasoning and Knowledge in Large Language Models. arXiv preprint arXiv:2407.12844.
>
> Q4: Comparison recovery error of raw scores.
>
> Q4-Ans:  Kipnis et al. (2025) fit one-dimensional IRT models on a dataset containing **more than 5,000 LLMs** evaluated on **six benchmarks**. Their data scale is substantially larger than ours and is therefore much more suitable for fitting complex IRT models.
> To further illustrate this point, we train a 1PL IRT model for 3000 epochs using the py-irt library  on our items, which contains:
>
> - 32 models,
> - 8 reasoning benchmarks,
> - a total of 2,506 problems,
>
> With the IRT model, we repeated the prediction-error experiment shown in Figure 6. At **known task num k = 7**, the IRT model exhibits a mean absolute error exceeding **20 percentage points**, which is substantially worse than the ≈ **4 points** error achieved by our MPP-based approach.
>
> | k | 1 | 2 | 3 | 4 | 5 | 6 | 7 |
> | --- | --- | --- | --- | --- | --- | --- | --- |
> | MAE (points) | 29.598 | 26.131 | 24.413 | 23.435 | 22.609 | 21.062 | 20.750 |
>
> Q5:  Why reasoning LLMs and benchmarks?
>
> Q5-Ans:  Our focus on reasoning models and reasoning benchmarks is intentional. The goal of this work is to isolate and analyze a **specific capability dimension**, namely, reasoning ability. This also helps maintain the mathematical simplicity of our model.
>
> Conceptually, the framework is agnostic to model type. However, we expect the method to perform best in scenarios where the evaluation focuses on a relatively homogeneous capability.
>
> We remain actively available throughout the rebuttal phase and are ready to provide further clarification.

---

> > ### Comment · Reviewer_EHvj · 2025-11-26
> >
> > Thank you for your engagement with my feedback, I appreciate the effort that has been made to remedy some of the issues with the paper. The further clarification of the methodology and tidiness of the presentation is useful. While I agree that IRT and your method are different, I think moving towards instance level analysis (as in the IRT case) is where we should be going with AI evaluation, rather than scoring and prediction on large, poorly constructed datasets. While it could be useful from an engineering/iteration stand point to be able to commensurate across heterogenous benchmarks, I don't think this work moves the needle towards better measurement practices in the science of evaluation. Therefore, I choose to retain my score.

---

> > > ### Author Response · Authors · 2025-12-01
> > > **Rebuttal by Authors**
> > >
> > > We sincerely thank you for the careful evaluation of our work and the constructive comments. We agree with your perspective that instance-level evaluation, such as IRT-style approaches, represents an important future direction for AI assessment.
> > >
> > > At the same time, we respectfully believe that our work provides a useful and novel step towards that shared goal. As pointed out by the reviewers, the proposed Model Power Parity (MPP) framework introduces several unique advantages to current evaluation methodologies:
> > >
> > > - First, MPP offers a **conceptually novel and rigorous formulation** grounded in well-established statistical foundations in PPP theory (as highlighted by Reviewer **Cade, onZw, QWye and EHvj**: “strong innovation”, “solid theoretical foundation”, “clear derivation and reproducibility”, “The idea behind this paper is novel”, “using well-established methods from macroeconomics”).
> > > - Second, it **models benchmark difficulty and model capability jointly**, without presuming a universal scale, which is a limitation present in previous score-based or normalized metrics. This enhances the interpretability of cross-benchmark comparisons (Reviewer **QWye**: “original and elegant analogy”, “enables transitive comparisons”).
> > > - Third, MPP demonstrates **practical utility and scalability**, validated across more than 30 open-source models and 8 mainstream reasoning benchmarks, supporting model ranking, performance prediction, and benchmark selection (Reviewer **onZw and EHvj**: “comprehensive experiments with convincing results”, “The fact that raw score can be recovered from the inferred PPP is a useful result”).
> > >
> > > While MPP does not yet directly operate at the instance-response level, it helps address heterogeneity and inconsistency across existing benchmarks (many of them lack item-level results), which current instance-level methods alone cannot fully solve.  Therefore, we believe that integrating the strengths of instance-level methods like IRT with the multilateral comparability advantages of MPP is a valuable future direction.
> > >
> > > Thank you again for your thoughtful review and for helping us improve the clarity and positioning of our contribution.

---

### Official Review · Reviewer_QWye · 2025-10-27

**Soundness:** 3
**Presentation:** 3
**Contribution:** 3
**Rating:** 6
**Confidence:** 3

**Summary:**

This paper proposes Model Power Parity (MPP) — a novel evaluation framework for comparing reasoning abilities of large language models (LLMs) across benchmarks of differing difficulty. Inspired by Purchasing Power Parity (PPP) in economics, MPP models the relationship between model capability and benchmark difficulty without assuming a shared difficulty scale.

**Strengths:**

1, Adapting PPP theory from economics to model–benchmark evaluation is an original and elegant analogy, connecting economic parity with model capability parity.
2, MPP does not assume a single global difficulty scale, unlike average or normalized benchmarks; it enables pairwise and transitive comparisons, enhancing interpretability.
3, Clear derivation using CPD-MTD and Jevons-GEKS-MTD formulations ensures reproducibility and links to established econometric theory.
4, Covers >30 open-source reasoning LLMs and 8 major benchmarks, providing a credible empirical foundation.

**Weaknesses:**

1， The CPD model assumes linear additive separability of “model ability” and “task difficulty.” Real LLM–benchmark interactions may be nonlinear and interaction-heavy, undermining this assumption.
2， No confidence intervals, variance estimates, or robustness checks are provided for MPP/TPP values — critical for establishing reliability.
3， Since both model and task parameters are estimated from the same performance matrix, there’s risk of self-normalization bias, especially with sparse or correlated data.

**Questions:**

-

**Details Of Ethics Concerns:**

-

---

> ### Author Response · Authors · 2025-11-22
> **Rebuttal by Authors**
>
> Thank you for the thoughtful and valuable feedback! We appreciate your comments and have carefully addressed each concern point by point below:
>
> W1: Real LLM–benchmark interactions may be nonlinear and interaction-heavy, undermining the CPD assumption.
>
> W1-Ans: We agree that the interaction between model capability and benchmark characteristics can, in principle, be nonlinear. However, our goal in this work is **not to fully model the cognitive process or response behavior of LLMs**, but to provide a **stable, interpretable, and scale-compatible way to adjust performance scores across heterogeneous benchmarks**.
>
> The linear additive structure consistent with the CPD formulation is chosen deliberately for two reasons:
>
> (1) **Domain-Constrained Evaluation**
>
> We focus specifically on **reasoning models and reasoning benchmarks**, rather than heterogeneous multimodal or multi-skill evaluation settings. By narrowing the scope to reasoning ability, we reduce the dimensionality of capability factors. In this domain-restricted setting, a one-factor additive model provides reasonable ranking results as shown in **Table 1.**
>
> (2) **Practicality and Interpretability**
>
> The available dataset with 32 models on 8 benchmarks is insufficient to reliably fit nonlinear or high-interaction models. In contrast, the additive formulation converges reliably and preserves interpretability.
>
> W2: Lack of confidence intervals or variance estimates for MPP.
>
> W2-Ans:  We have supplemented the formulas for calculating the variance of PPP in the CPD method, which are presented in **Equations 7 to 9 of Section 2.2.** The MPP computation can be written explicitly for variance estimation: **$Score_{mt} = \alpha + U_m + V_t + u_{mt}, u_{mt} \sim \mathcal{N}(0, \sigma^2).$**
>
> We select some representative models and present the variance values and confidence intervals of their MPP as follows:
> | Model                         | MPP| Variance | 95% CI           |
> | ----------------------------- | -------------: | -------: | ---------------- |
> | DeepSeek-R1-Distill-Qwen-1.5B |0.6214 | 0.0529 | [0.1708, 1.0720] |
> | DeepSeek-R1-Distill-Qwen-7B   |1.4663 |0.2944 | [0.4029, 2.5297] |
> | DeepSeek-R1-Distill-Llama-8B  |1.1332 |0.1758 | [0.3114, 1.9550] |
> | DeepSeek-R1-Distill-Qwen-14B  |     2.5311 |   0.8770 | [0.6955, 4.3666] |
> | DeepSeek-R1-Distill-Qwen-32B  |     2.8490 |   1.1112 | [0.7829, 4.9151] |
> | Qwen3-4B                      |     2.7415 |   1.0289 | [0.7534, 4.7296] |
> | Qwen3-8B                      |     3.2660 |   1.4603 | [0.8975, 5.6346] |
> | Qwen3-14B                     |     3.7864 |   1.9627 | [1.0405, 6.5323] |
> | Qwen3-32B                     |     4.2119|   2.4286 | [1.1574, 7.2663] |
>
> The results show that models with **more homogeneous performance** across tasks exhibit smaller estimated variances and narrower confidence intervals. Although some intervals may appear wide, they are proportional to model scaling and task variability, and importantly, the ordering implied by the point estimation **is preserved** even after considering uncertainty.
>
> W3: Risk of self-normalization bias, especially with sparse or correlated data.
>
> W3-Ans: Self-normalization bias is indeed a recognized characteristic of CPD-style multilateral index methods, rather than a flaw specific to MPP. Similar to PPP estimation, the estimated scale is internally normalized based on overlapping observations (Rao 2013), and prediction uncertainty increases when score sparsity reduces the statistical overlap between models and benchmarks. We observe this behavior in our experiments (Figure 6), and we explicitly discuss it in the limitation section. However, despite the sparsity, the MPP framework still **maintains stable relative ordering among models and produces statistically meaningful and interpretable capability estimates**, as demonstrated in **Table 1**. This suggests that, while normalization effects exist, the resulting capability scale remains consistent enough for practical comparison and aggregation across benchmarks.
>
> We view this not as a limitation unique to our framework, but as an inherent trade-off shared by most latent-factor estimation methods under sparse supervision. In future work, we plan to explore regularization strategies or weighted constraints to further mitigate normalization effects.
>
> Dodla Sai Prasada Rao. Computation of basic heading ppps for comparisons within and between regions. 2013. URL https://api.semanticscholar.org/CorpusID:150615236.

---

### Official Review · Reviewer_onZw · 2025-10-30

**Soundness:** 3
**Presentation:** 3
**Contribution:** 3
**Rating:** 6
**Confidence:** 4

**Summary:**

A new evaluation framework called Model Power Parity (MPP) has been proposed. This framework considers models as "currency" and benchmark tests as "commodities", using a multilateral comparison method to estimate both the model's ability and the difficulty of the benchmark, without the need to preset a uniform difficulty scale. The author conducted experiments on over 30 open-source inference language models and 8 mainstream inference benchmarks to verify the effectiveness of MPP in model ranking, performance prediction, and benchmark selection.

**Strengths:**

1: Strong innovation: Introducing PPP ideas from economics into LLM evaluation, with a novel perspective and inspiring methods.

2: Solid technology and rigorous methodology: Two specific calculation methods (CPD-MTD and JEVSON-GEKS-MTD) are provided, both based on standard statistical methods in the PPP field, with a solid theoretical foundation. Reasonable preprocessing (log Odds conversion) was performed on the scores, handling different indicators such as probability and perplexity. The experimental design is systematic, covering multiple dimensions such as ranking, prediction, and correlation analysis.

3: The experiment is comprehensive, covering multiple mainstream open-source models (such as Qwen3, DeepSeek-R1, Phi-4, etc.) and multiple inference benchmarks, with a large amount of data and convincing results.

**Weaknesses:**

1: Limited benchmark scope: Only 8 inference class benchmarks were used, covering limited domains and task types, which may affect the framework's generalization ability.

2: The model assumption is relatively simple: the current model assumption is that performance is a linear sum of model capability and task difficulty, without considering more complex interaction effects (such as model task adaptability).

3: Lack of comparison with existing methods: No systematic comparison with existing benchmark difficulty assessment methods.

4: The theoretical explanation for why PPP is effective is slightly weak: the paper provides a good description of how to do it, but the theoretical connections and assumptions behind why the analogy of PPP is effective in LLM evaluation can be further elaborated.

5: The readability of some charts is average: for example, the explanations of Figure 3 and Figure 5 are not clear enough, and the reader's understanding cost is high.

**Questions:**

1: Is the MPP framework suitable for non-inference tasks such as generation, dialogue, and multimodality, and does it need to be adjusted?

2: Is there a non-linear relationship between modeling ability and task difficulty, and are there plans to introduce more complex modeling methods?

3: Is there a plan to quantitatively compare MPP with existing benchmark difficulty assessment methods to further highlight the advantages of MPP?

4: Are there any models or task types that are more prone to significant errors in performance prediction, and are there any patterns to follow?

5: Have you considered extending MPP to dynamic evaluation scenarios?

---

> ### Author Response · Authors · 2025-11-22
> **Rebuttal by Authors**
>
> Thank you so much for your constructive and insightful comments! We address your concerns point by point as follows:
>
> Q1: The adaptability of MPP framework for non-inference tasks?
>
> Q1-Ans: For the non-inference tasks you mentioned (e.g., generation and dialogue), the MPP framework can be **directly applied** as long as specific score metrics of models on domain-specific benchmarks are obtainable, with only the need to replace them with the corresponding models and select appropriate benchmarks.
>
> Q2: Relationship between modeling ability and task difficulty, and plans to introduce more complex modeling methods?
>
> Q2-Ans: We agree that the relationship between model ability and task difficulty is likely not a simple linear one. However, due to the relatively small scale of the data used for evaluation, **a nonlinear model may not converge well**. The simple additive structure we used manages to capture the primary sources of variation, offering good interpretability and reducing concerns about model convergence, which is also a design commonly adopted in econometrics. As shown in **Table 1**, the MPP method produces model capability rankings that **align with general expectations**, which suggests that **the linear assumption does not mislead the framework**. As discussed in the future work part, we are interested in exploring formulations where different benchmark categories contribute with different weights, as well as integration with methods like Item Response Theory when rich item-level data is available. These directions may allow MPP to capture richer model–task relationships while maintaining its core benefit of **providing a unified and comparable evaluation scale.**
>
> Q3: Quantitative comparison with existing benchmark difficulty assessment methods?
>
> Q3-Ans:  We agree that a quantitative comparison between MPP and existing benchmark difficulty estimation works would help further validate the methodology. Prior works as we mention in the **Introduction section**,  operate at the **item level** (e.g., difficulty scoring, human annotation, or rule-based hardness tagging), whereas MPP is designed to work at the **benchmark-level aggregation layer**, with a focus on enabling cross-domain comparability and prediction rather than labeling individual test items. This mismatch in granularity makes a controlled quantitative comparison nontrivial. To be specific, difficulty labels in prior work (e.g., Rein et al., 2023) are often domain-specific or rely on human annotation, making them difficult to align to a standardized numeric scale for direct statistical comparison with MPP-derived difficulty estimates.
>
> Although a large scale quantitative comparison would require costly consistent item-level data, we view this direction as valuable future work as benchmark ecosystem gets more fine-grained.
>
> David Rein, Betty Li Hou, Asa Cooper Stickland, Jackson Petty, Richard Yuanzhe Pang, Julien Dirani, Julian Michael, and Samuel R Bowman. Gpqa: A graduate-level google-proof q&a benchmark. arXiv preprint arXiv:2311.12022, 2023.
>
> Q4: Performance prediction patterns?
>
> Q4-Ans:  Yes. Our results indicate that certain task types are more prone to producing larger prediction errors, and the errors are not uniformly distributed across benchmarks as discussed in **Section 3.2**.
>
> Specifically, our cross-benchmark analysis (Figure 4) shows that prediction errors are structured rather than random. Benchmarks such as **MMLU-Pro** and **GPQA-Diamond** exhibit larger mutual prediction errors with mathematical competitions (e.g., AIME-2025), while their mutual prediction errors remain relatively small. This suggests that these benchmarks rely on partially different reasoning capabilities, which limits cross-predictability. In contrast, **MATH-500** demonstrates consistently low prediction error with nearly all other benchmarks, indicating that it shares overlapping reasoning skill requirements with multiple tasks. This observation is further supported by the k = 7 setting (Figure 5), where MATH-500 and GSM8K show the smallest average prediction error.
>
> Overall, the pattern indicates that prediction uncertainty increases when benchmarks measure distinct cognitive skills, or when model performance distribution is highly heterogeneous, whereas tasks with shared reasoning structure or high inter-model consistency yield more reliable predictions.
>
> Q5:  Extending MPP to dynamic evaluation scenarios?
>
> Q5-Ans: Currently we focus on a static, cross-domain evaluation, where each model is treated as a fixed entity with a single ability parameter. However, the structure of the method suggests that such an extension is feasible. Specifically, if a model produces outputs across multiple time points, we may treat each model-time pair as a distinct entity,  and recover a trajectory of ability parameters $U_m(t)$. We consider this an interesting direction and may explore it in future work.

---

### Official Review · Reviewer_Cade · 2025-10-31

**Soundness:** 1
**Presentation:** 1
**Contribution:** 2
**Rating:** 2
**Confidence:** 3

**Summary:**

The authors propose an analogy, within the ICP (and PPP) economics model, between currencies and LLMs and goods as benchmarks. In other words, LLMs are the currency that can buy benchmarks. In this sense, one can measure the PPP of an LLM to compare it to others. The higher the PPP, the more powerful a model is at reasoning.

**Strengths:**

I applaud the authors for trying to propose an unorthodox way to measure the performance of an LLM and compare it to others using well-established methods from macroeconomics.
There may be some value in this approach if expanded and actually grounded to concepts where the analogy makes sense.
With that being said, please read the weaknesses.

**Weaknesses:**

The framework is very confusing, and it is not clear, from the very beginning, what the actual analogy the authors are posing. If the models are the currencies, who owns and how are the currencies spent? The paper jumps, right after proposing this analogy without giving insights, into the mathematical framework, which is poorly written and exposed. For example, what is p in Eq. 1?

To make it clear, my main concern is with the entire analogy of the paper: a good is something that provides a utility and is scarce. I cannot see the analogy with benchmarks, which are not scarce (I cannot have 100 MATH500, but I can have infinitely many copies of MATH500).
When it comes to currencies, they have 3 key functions: medium of exchange (for trading goods), store of value (they measure the price of things) and unit of account (for savings).
I cannot really see the analogy with LLMs. In what sense do they serve as the currency for benchmarks? Currencies are also fungible, but LLMs are not: I cannot consider 100 tokens of Llama 4 on MATH500 as 100 tokens of GPT-4o on GSM8K; they are different things, and the analogy here is misleading on every aspect, without even mentioning the other aspects of a currency (portable, divisible), which do not hold in the framework.

In terms of results, Table 2 has many missing values (the caption claims every model has a predicted and real score).

The entire paper is built around a framework that, if I am not wrong, could be stated as “the more benchmarks a model can solve in a category and in general, the better it is at reasoning”. But that is precisely what people do in benchmarking, without the necessity to use this framework. One can measure (and researchers do, check HELM by Stanford) the performance on a dataset, as well as the (weighted) average performance on a class/category (e.g., math).

**Questions:**

Q1. What does it mean for a model (the currency in the PPP framework) to buy a dataset?
I ask the reviewers to clarify the analogy between LLMs, datasets, and the PPP framework; for me, that is not clear at all, and the analogy misplaced.
I elaborate further on my concerns in the section Weaknesses.

Q2. Table 2 has missing values (the authors say each result consists of two numbers, but only a few have both). Is it intended to be so?

Q3. Can the authors elaborate on the difference and advantages of this framework compared to standard benchmarking (works like HELM or more recent papers on benchmarking models per-class/category of datasets)?

---

> ### Author Response · Authors · 2025-11-21
> **Rebuttal by Authors**
>
> Thank you for your constructive and insightful comments. Below, we first clarify the overall analogy, and then we will address each of your questions one by one. To make our MPP framework easier to understand, we draw **an overview fig (Figure 1)** in the revised pdf version. We also revise the equations to in the method section for better understanding.
>
> **We would like to clarify the analogy here:** although our framework is inspired by economic concept, it does not aim to establish a strict economic model. Based on this concept, we draw an analogy: **as different countries use different currencies whose values cannot be directly compared without considering price levels, different LLMs produce performance scores on different benchmarks that are not directly comparable due to varying task difficulty. In the same way that a currency with higher purchasing power can buy more goods in different countries, a stronger model can “purchase” higher scores across benchmarks with different difficulty levels.**
>
> We jointly solve for the model capability and benchmark difficulty through regression to evaluate Model Power Parity (MPP) and Task Power Parity (TPP), similar to how the Country Product Dummy (CPD) method jointly estimates the international average price and the country-specific commodity price effect through regression to calculate PPP. The analogy is meant to help readers intuitively understand the structural decomposition, but it does not imply that LLMs possess economic attributes such as scarcity, substitutability, or portability in the way that currency does.
> We take the formulas as an example:
> In the CPD formula: **$\ln(p_{ij}) = \ln(PPP_j) + \ln(P_i) + \ln(u_{ij}) = \alpha_i + \gamma_i + \nu_{ij},$**
>
> $\alpha_i$ represents the country-specific commodity price effect, $\gamma_i$  represents the international average price, $\nu_{ij}$ represents independently and identically distributed random disturbances.
>
> And in our MPP formula: **$Score_{mt} = \alpha + U_m + V_t,$**
>
> $U_m$ represents model capability, $V_t$ represents benchmark difficulty.
>
> Q1：What is the analogy between MPP and PPP indeed?
>
> Q1-Ans: We describe the analogy above. We revise the paper pdf version to include more details about how we build our math model from the original PPP theory in the **method section**.
>
> Q2: Table 2 has missing values (the authors say each result consists of two numbers, but only a few have both). Is it intended to be so?
>
> Q2-Ans: The values in parentheses represent predicted performance, while the values without parentheses are actual performance. Entries that include both predicted and actual performance represent the assumption that the score is missing and will be predicted. For example, “85.60 (85.16)” means that the actual performance is 85.60 and the predicted performance is 85.16. **The purpose of Table 2 is to demonstrate that our framework can predict benchmark scores using only single-value entries, without the need to run inference on the benchmark.**
>
> Q3: The difference and advantages of this framework compared to recent standard benchmarking works.
>
> Q3-Ans: Standard benchmarking frameworks such as **HELM** primarily focus on **reporting or organizing raw performance metrics** across tasks. These works summarize how well models perform on a collection of benchmarks respectively, but they typically do not provide a way to **normalize or adjust scores across heterogeneous domains**.
>
> In contrast, our work addresses a different problem:
>
> **How can we place heterogeneous benchmarks onto a shared difficulty scale, such that scores across domains become comparable and can be interpreted meaningfully?**
>
> | Aspect | Standard Benchmarking  | MPP Framework |
> | --- | --- | --- |
> | **Primary Goal** | Report and structure evaluation across tasks | Estimate **relative benchmark difficulty** and **normalize model performance** across tasks |
> | **Assumption** | Scores are directly comparable or reported separately | Scores require **difficulty adjustment**, similar to PPP in economics |
> | **Granularity** | Aggregation or per-category analysis | Cross-domain statistical calibration |
> | **Output** | Raw scores and descriptive comparisons | Difficulty-adjusted model capability (MPP) and benchmark difficulty (TPP) |
> | **Scalability to new benchmarks** | Requires new evaluation | Allows extrapolation and score prediction  |
>
> To sum up, we view MPP as **helpful complementary rather than an alternative** to works like HELM:
>
> - HELM organizes what to measure.
> - MPP provides a principled method for *how to compare* results across benchmarks with unequal difficulty.
>
> We sincerely appreciate your time and constructive feedback. We remain actively available and committed during the rebuttal period to clarify any remaining concerns and further improve the quality of the work.

---

### Author Response · Authors · 2025-12-02
**Summary of Reviews and Rebuttal Process**

To ACs:

Thanks for your service to ICLR26! We respectfully submit this brief summary to assist you in evaluating our paper.

## 1. Summary of our paper
**Goal:** We establish a principled and interpretable framework for evaluating LLM capabilities, addressing the challenges of LLM  ability comparison on cross-domain benchmarks.

**Core idea:**

To the best of our knowledge, we are the first to adapt Purchasing Power Parity (PPP) theory from economics to LLM evaluation by introducing the Model Power Parity (MPP) framework for multilateral comparison across models and benchmarks.

**Results:**
1. Ranking validity: MPP yields stable, intuitive model rankings consistent with reasonable capability differences.
2. Score recovery: Achieves approximately 0.5–3 point error on the two benchmarks exhibiting the highest recovery accuracy.
3. Benchmark selection insight: Reveals which benchmarks are misleading or unsuitable for specific model families.

**Strengths:**

Reviewers generally agree that:
1. Novelty of the core idea: creatively adapting PPP methodology to LLM benchmarking.
2. Methodological soundness: rigorous grounding based on established econometric formulations.
3. Interpretability and useful properties: including recoverability of raw scores and meaningful benchmark difficulty estimation.

## 2. Reviews Overview
At the initial state, we received scores of **6 / 6 / 2 / 2** with confidence **4 / 3 / 3 / 3**. Until now, the only reviewer who responded to our rebuttal is Reviewer EHvj (with rating 2). The other three reviewers **have not responded** since we posted the rebuttals on November 22.

## 3. Response to Reviewer Cade (Rating: 2)
The main concerns are 3 points:
1. How the analogy between “models as currencies” and “benchmarks as goods” is formulated and interpreted.
2. Writing clarity.
3. How MPP compares to standard benchmarking works.

**Our actions:**
1. We provided a clearer conceptual grounding of the model ability–benchmark difficulty relationship through the currencies–goods analogy and revised the manuscript in Paragraph 4 of Section 1 Introduction.
2. We revised the manuscript in Figure 1 and Section 2.1 of Methodology as suggested.
3. We further included a dedicated comparison table that explicitly contrasts MPP with standard benchmarking works like Holistic Evaluation of Language Models (HELM), making clear where our framework provides additional functionality: HELM organizes what to measure, while MPP provides a method for *how to compare* results across benchmarks with unequal difficulty.

## 4.Response to Reviewer EHvj (Rating: 2):
The main concerns are 5 points:
1. Practical benefit of MPP beyond ranking recovery.
2. Multicollinearity in the Country Product Dummy (CPD) regression.
3. Comparison to instance-level methods like Item Response Theory (IRT).
4. Score recovery accuracy comparison.
5. Reason for choosing reasoning models.

**Our actions:**
1. We clarified the unique value of MPP in enabling cross-benchmark comparability and performance prediction on unseen benchmarks.
2. We explained how reference constraints mitigate collinearity and revised the manuscript in Section 2.2 of Methodology.
3. We added a clear comparison on motivation and data source respectively to show the difference between MPP and IRT.
4. We provided new quantitative comparison experiments demonstrating that MPP outperforms IRT model under current settings.
5. We further clarified how this choice suits our current mathematical model.

**Reviewer EHvj's Feedback:**

The reviewer noted meaningful improvements after addressing earlier concerns. He/she explicitly recognized that our **methodology has been clarified** and the **presentation has become more organized and precise**. He/she also reaffirmed that **IRT and MPP are conceptually distinct, and** also acknowledged the **practical usefulness** of MPP from an **engineering and model iteration standpoint**.

**Our Follow-up:**

This feedback helped us more clearly articulate the complementary strengths of IRT approaches and our MPP framework, which also highlighted **a valuable future direction:** integrating the item-level diagnostic strengths of IRT with the multilateral comparability advantages offered by MPP.

## 5.Response to Reviewer onZw (Rating: 6) and QWye (Rating: 6):
Besides the same concerns mentioned above, they had some minor concerns, such as the generalization of MPP, more statistical analysis and future work. We also gave detailed explanations and revised the manuscript as suggested.

## 6. Conclusion
We sincerely thank all the reviewers and ACs for improving our paper. The comments have been well taken and reflected in our revised paper (see the highlighted content in blue in the PDF).

We believe the new experiments and clarifications resolve the reviewers' concerns and help better convey the contributions of our work.

Thanks for your precious time, ACs!

Best,

Authors

---

### Meta-Review · Area_Chair_hUoe · 2026-01-04

**Summary:**

The authors propose a new way to measure the performances of LLMs on varoius benchmarks with diffferent difficulty levels. This work may have some unrealized potential, but its current form is far from acceptable for ICLR. The economical analogy is very confusing to multiple reviewers (and myself), even after rebuttal. In addition, the lack of systematic comparison with existing benchmark difficulty assessment methods is another major issue. The scores are highly diverse: two 6s and two 2s. I think this reflects the potential of the paper, but also highlights the unreadiness of the paper.

**Reviewer Concerns:**

The main concerns are regarding the suitability of the ICP analogy and its corresponding proposed MPP, as well as comparison with the existing benchmark difficulty assessment methods. These major issues are largely unresolved based on the rebuttal, in my opinion,.

**Reviewer Scores:**

I can see some minor changes of the score (2->3), but overall not significant.

---

### Decision · Program_Chairs · 2026-01-26

Reject